# Chemical Compositions, Pharmacological Properties and Medicinal Effects of Genus *Passiflora* L.: A Review

**DOI:** 10.3390/plants13020228

**Published:** 2024-01-13

**Authors:** Krastena Nikolova, Margarita Velikova, Galia Gentscheva, Anelia Gerasimova, Pavlo Slavov, Nikolay Harbaliev, Lubomir Makedonski, Dragomira Buhalova, Nadezhda Petkova, Anna Gavrilova

**Affiliations:** 1Department of Physics and Biophysics, Medical University-Varna, 9000 Varna, Bulgaria; 2Department of Physiology, Medical University-Varna, 9000 Varna, Bulgaria; msvelikova@yahoo.com; 3Department of Chemistry and Biochemistry, Medical University-Pleven, 5800 Pleven, Bulgaria; 4Department of Chemistry, Medical University-Varna, 9000 Varna, Bulgaria; anelia.gerasimova@mu-varna.bg (A.G.); lubomir60@yahoo.com (L.M.); 5Faculty of Medicine, Medical University-Varna, 9000 Varna, Bulgaria; pavloslavov@gmail.com (P.S.);; 6Department of Nutrient and Catering, University of Food Technology, 4002 Plovdiv, Bulgaria; dra.buhalova@gmail.com; 7Department of Organic Chemistry and Inorganic Chemistry, University of Food Technology, 4002 Plovdiv, Bulgaria; nadezhda_petkova@uft-plovdiv.bg; 8Department of Pharmaceutical Chemistry and Pharmacognosy, Medical University-Pleven, 5800 Pleven, Bulgaria; any_gavrilova@abv.bg

**Keywords:** *Passiflora*, medical application, food supplement, chemical composition, cosmetics, safety, toxicity

## Abstract

Practically all aboveground plants parts of *Passiflora* vines can be included in the compositions of dietary supplements, medicines, and cosmetics. It has a diverse chemical composition and a wide range of biologically active components that determine its diverse pharmacological properties. Studies related to the chemical composition of the plant are summarized here, and attention has been paid to various medical applications—(1) anti-inflammatory, nephroprotective; (2) anti-depressant; (3) antidiabetic; (4) hepatoprotective; (5) antibacterial and antifungal; and (6) antipyretic and other. This review includes studies on the safety, synergistic effects, and toxicity that may occur with the use of various dietary supplements based on it. Attention has been drawn to its application in cosmetics and to patented products containing passionflower.

## 1. Introduction

The majority of species of the genus *Passiflora* L. are vines [1]. The fruits of most species are used for the preparation of juices, jams, and soft drinks. The byproducts isolated from the fruit waste products and pulp are included in the compositions of dietary supplements, pharmaceuticals, and food products [2].

The *Passifloraceae* family consists of 16 genera, including more than 600 species [3]. The genus Passiflora alone contains more than 450 species, most of them indigenous to the tropics and subtropics of the New World. Most *Passiflora* species are used in the pharmaceutical and food industries, cosmetics, and more. The most common species are *Passiflora edulis* Sims (purple passion fruit), *Passiflora edulis* f. *flavicarpa* O. Deg (yellow passion fruit), *Passiflora ligularis* Juss. (sweet granadilla), *Passiflora tarminiana* Coppens & V. E. Barney (banana passion fruit), *Passiflora nitida* Kunth (bell apple), *Passiflora setacea* DC. (sururuca), *Passiflora incarnata* L., *Passiflora quadrangularis* L. (giant granadilla), and ‘Kaveri’ Hybrid passion fruit (purple and yellow) [3]. The species distributions and fruit descriptions of some *Passiflora* representatives are listed in Table 1.

The rich chemical composition of *Passiflora* species predetermines their diverse applications in treating insomnia [7], anxiety [8], epilepsy, asthma [9], and high blood pressure [10]. Among the *Passiflora* species, *P. edulis* and *P. incarnata* are the most frequently used ones in pharmacology and medicine.

Several pharmacological studies indicate that various bioactive components, such as polyphenols, triterpenes, carotenoids, polysaccharides, amino acids, essential oils, and flavonoids, contribute to the antioxidant, antimicrobial, antidiabetic, hepatoprotective, and anti-inflammatory effects of *Passiflora* species [11,12,13,14,15,16]. Besides their medicinal applications, the fruits of certain species are utilized in the food industry due to their richness in dietary fiber, vitamins, minerals, and other beneficial compounds [17].

The purpose of this review is to systematize the knowledge about the chemical composition and biological activity of the different plant parts of *Passiflora* species and relate it to their applications in medicine and cosmetics.

## 2. Taxonomy

The taxonomic classification of the *Passiflora* L. [18] genus and Figure 1, showing the flowers and fruits of some *Passiflora* species, are below:

Kingdom: *Plantae*.

Phylum: *Tracheophyta.*

Subphylum: *Spermatophytina.*

Class: *Magnoliopsida.*

Superorder: *Rosanae.*

Order: *Malpighiales.*

Family: *Passifloraceae.*

Genus: *Passiflora.*

Due to the extensive species diversity in the *Passiflora* genus, its generalized and at the same time unambiguous description can be challenging. In summary, most *Passiflora* species are lianas, vines, or herbaceous plants, mostly climbing, with axillary tendrils. Nevertheless, there are some tree species as well. The stems are cylindrical, angular, or, less often, winged. All species have well-developed stipules. Leaves are alternate, simple, entire, lobed, or palmate, rarely compound. A distinctive feature of the genus is the extrafloral nectarines. The flowers are axillary, regular, and hermaphroditic. The perianth has five sepals and five petals, often similar in color and shape. Its color varies from flat green to violet or red. Maybe the most characteristic floral feature of the *Passiflora* genus is the centrally positioned filamentous corona with circular flower nectar at the base. The famous edible fruits of the genus are yellow or orange-to-red capsules with many seeds.

## 3. Chemical Composition

Polyphenols are one of the main classes of compounds in *Passiflora* species. They are associated with the color and astringent taste of the drinks prepared from the fruits [19]. According to Wen et al., the peel of *P. edulis* and *P. edulis* f. *flavicarpa* also contains fiber (22.5%), pectin (12.5%), and polysaccharides (20.62%) [20]. The insoluble dietary fiber in *P. edulis* seeds is over 64.1%. In addition, the fruits are rich in some flavonoids, such as apigenin-3-rhamnoside (4.01 mg/100 g), luteolin-3-glucoside (3.63 mg/100 g), quercetin (2.21 mg/100 g), and kaempferol (1.78 mg/100 g) [21]. The flavonoids might contribute to the antiallergic, antitumor, antiviral, and neuroprotective effects of *P. edulis* [22]. These compounds act as scavengers of oxygen-reactive species and reduce oxidative stress [23]. There is evidence that flavonoids can cross the blood–brain barrier and possibly have a positive effect on the prevention and treatment of neurodegenerative diseases [24].

The flavonoids apigenin and chrysin in *P. incarnata* are associated with a sedative effect [25]. Additionally, flavonoids can act by protecting DNA against oxidative damage and possess antiviral, antiallergic, cytotoxic, and therapeutic properties [26,27]. The classification of flavonoid types and their contents in the fruit and peel of different Passiflora species are presented in Figure 2.

Information on the chemical compositions of different *Passiflora* species is presented in Table 2.

Phenolic compounds are not only present in the pulp, peel, and fruits of *Passiflora*, but also in the leaves, which are rich in flavonoids such as apigenin, luteolin, quercetin, and kaempferol [43,44,45]. In addition, *Passiflora* fruits are also rich in nitrogen, potassium, calcium, and phosphorus. For example, yellow passion fruit contains 2400 mg of nitrogen per 100 g of pulp [46]. Potassium is essential for cell functions, nerve impulse transmission, and muscle contractions, and its content in the fruits varies among the separate species, from 1440 to 3800 mg per 100 g of pulp [47,48,49].

*Passiflora* species differ in mineral content and antioxidant activity, with *P. setacea* being high in potassium, calcium, magnesium, and more [50]. *P. nitida* stands out for its iron content and high antioxidant potential [51,52,53,54]. Iron is an essential element involved in various metabolic processes, including oxygen transport in blood and electron transport in the mitochondria [46,55].

Some species, like *P. edulis* and *P. ligularis*, are rich in zinc (Zn), but in addition to Zn *P. ligularis* also contains high concentrations of copper (Cu) [46,49,56]. Therefore, the consumption of *P. ligularis* should be encouraged, which can be used as a micronutrient supplement to improve natural immunity against COVID-19 and its new variants [57]. In a small number of sources there are also data on the mineral compositions of the leaves of the plant, since they are mainly used for different types of extracts (water, alcohol, or water–alcohol). According to Freitas et al., *P. edulis* f. *flavicarpa* leaves are rich in calcium and zinc, with their content strongly influenced by the growing season. In young leaves, the levels of Zn, N, and P are high, while those of Ca, Mg, B, Cl, and Mn are relatively low [58]. In addition to minerals, passion fruit is also rich in vitamins, containing vitamin C (30.7 mg/100 g) and high levels of provitamin A (233.2 μg/100 g) [59]. Table 3 presents the vitamin compositions of juices prepared from some types of passionflower.

*Passiflora* fruits are also rich in dietary fiber, including carbohydrate polymers such as oligosaccharides, pectin-type polysaccharides, inulin, cellulose, and more. The fruit is the main source of fiber, and several studies have revealed its potential use as an ingredient in the food industry. Figure 3 summarizes studies on fiber content in *Passiflora* species, and Table 4 presents the chemical structures of specific compounds found in *Passiflora* as well as their pharmacological effects.

Ethanolic extracts from *Passiflora* leaves exhibit sedative and antidepressant effects attributed to the presence of cycloartane triterpenoid saponins, specifically cyclopacifloside XII and cyclopacifloside XIII [61]. Various parts of many *Passiflora* species are rich in amino acids and flavonoids. For example, the pulp of *P. edulis f. flavicarpa* is abundant in both essential and non-essential amino acids (11.88–12.21 g/100 g pulp). Arginine, glutamic acid, and aspartic acid collectively make up approximately 50% of all amino acids [62]. The total amino acid content varies among different *Passiflora* species; for instance, J. R. Souza et al. reported that *P. alata* contains 10.46 mg/g of amino acids, *P. edulis* f. *flavicarpa* has 7.74 mg/g, and *P. incarnata* contains 5.98 mg/g. [63]. *P. edulis* fruits and leaves contain alkaloids such as harmidine, harmine, harmane, harmol, *N-trans*-feruloyltyramine, and *cis-N*-feruloyl tyramine. These alkaloids inhibit monoamine oxidase A, suppress tumor growth and progression, and also inhibit the NF-kB signaling pathway, leading to anti-inflammatory effects [15,64,65].

The *Passiflora* fruit is highly acidic, with a pH level around 3.0 due to the presence of citric and malic acids [65]. Purple-variety fruits are typically consumed raw, while yellow-variety fruits are used to make soft drinks, juice concentrates, ice creams, and jams [4]. Sugar content varies among different *Passiflora* species, with yellow and purple fruits having higher sugar contents than more sour species. [4]. The primary sugars found in *Passiflora* fruit pulp are glucose, fructose, and sucrose [4]. Sema and Maiti [66] compared the total sugar content of three *Passiflora* species. The sugar contents of ripe fruits of some *Passiflora* species passion fruit are presented in Figure 4 [66]. The total content of major nutrients [4] in ripe fruits of some *Passiflora* species is presented in Figure 5.

## 4. Medical Effects and Activities in Pharmacy

Plants belonging to the *Passiflora* genus have a long history of use in traditional medicine for various health benefits, some of which are presented in Figure 6. Extracts from all plant parts of *Passiflora*, including fruit, bud, peel, and leaf, have been examined for antioxidant [67], antibacterial [68], anti-inflammatory [69], hepatoprotective, nephroprotective [70], antifungal [71], anti-fatigue, and other activities [72]. Evidence from experimental, preclinical, and clinical studies supports the hypnotic, anxiolytic, anti-depressant, anticonvulsant, and neuroprotective effects of *Passiflora* species.

### 4.1. Sedative and Anxiolytic Activities

Regarding sedative and anxiolytic effects, clinical studies have shown that *P. incarnata* extract can reduce tension, restlessness, and nervous irritability, making it beneficial for alleviating insomnia and anxiety. *Passiflora* extract has been compared to conventional anxiolytic drugs in various studies, suggesting that it may serve as an alternative or adjunct treatment option [73]. In recent years, the rates of depressive and anxious symptoms have increased due to COVID-19 [74], as well as the administration of antidepressants and antianxiety medications; however, many side effects are associated with their use, such as dependence and sexual problems. According to R. J. Bloomer et al., taking passionflower extract increased free salivary testosterone by 17% in men over 55 years of age and libido by 9%. Similar results were not observed in the group of examined men of about 35 years of age. According to the research team, the treatment is effective in individuals with low baseline testosterone levels [75]; therefore, phytotherapeutic preparations are increasingly recommended [76]. In a clinical study, it was found that an intake of 45 drops per day of *P. incarnata* extract for 28 days produces effects similar to those of oxazepam (a benzodiazepine) [77]. In a double-blind placebo-controlled study, orally administered as a premedication, *P. incarnata* reduced pre-operative anxiety with a similar effect to oral oxazepam [78]. In another study, the oral preoperative administration of *P. incarnata* suppressed anxiety before spinal anesthesia without changing psychomotor functions, sedation level, or hemodynamics [79]. In addition, it was reported that *P. incarnata* reduces dental anxiety in patients undergoing periodontal treatments [80]. These studies suggest that *Passiflora* extract may be a useful alternative or adjuvant to conventional anxiolytic drugs. A double-blind, placebo-controlled investigation on healthy adults with mild sleep disturbances reveals the beneficial effects on sleep quality of the low-dose consumption of *P. incarnata* in the form of tea [81]. The dysfunction of the GABA system is implicated in anxiety and depressive disorders. The pharmacological effects of *P. incarnata* are thought to be mediated via the modulation of the GABA system, including affinity for GABA(A) and GABA(B) receptors as well as GABA uptake [82]. Additionally, *Passiflora* extract has shown beneficial effects on some neurological disorders, although further research is needed to determine the specific parts of the plant used in these studies and the optimal dosages.

### 4.2. Anti-Depressant Activities

The forced swimming test (FST) in mice is commonly used as a behavioral test in assessing the antidepressant activity of substances. A study reported acute antidepressant-like effects of the aqueous extracts of *P. edulis* f. *flavicarpa* (1000 mg/kg, p.o.) and *P. edulis* f. *edulis* (300 mg/kg, p.o.) applied to mice in the FST [83]. The antidepressant activities of both *P. edulis* extract and extract-loaded nanoparticles were demonstrated in mice using the FST, with a 10-fold higher potency of the nanoparticles [84].

A study suggested that the products of the metabolism of *P. edulis*, possibly flavonoids, available in aqueous extract, and its fractions, butanol (BuOH, 25–50 mg/kg, p.o.) and ethyl acetate (AcOEt, 25–50 mg/kg, p.o.), are promoting antidepressant-like actions in mice. Evidence is presented that the antidepressant-like effect of the fractions may be dependent on monoaminergic neurotransmission [85].

### 4.3. Antidiabetic Activities

The antidiabetic effects of different parts of *P. edulis* have been investigated. In vivo experiments with rats showed that the 4 h consumption of 50 and 100 mg/kg leaf extract from *Passiflora suberosa* L. (cork-barked passionflower) produced a significant hypoglycemic effect. Fasting blood glucose levels decreased by 18%; following an oral sucrose challenge, intestinal glucose absorption was inhibited by 79%. The levels of total cholesterol decreased by 17%, and that of tri-glyceraldehydes decreased by 12% in the treated groups. The results suggest that the leaves of *P. suberosa* can be used to manage blood glucose and cholesterol levels [18]. The oral administration of *P. edulis* peel and seed extract for 15 days improved blood glucose levels in diabetic rats in a model of streptozotocin-induced oxidative stress [86]. Stilbenes such as scirpusin B and piceatannol, separated from the seeds of *P. edulis*, exhibited α-glucosidase inhibitory activity in vitro [87].

A randomized, placebo-controlled study was conducted on 39 subjects to evaluate the effects of piceatannol (20 mg/day), extracted from the seeds of *P. edulis*, on metabolic health. Supplementation with piceatannol reduced serum insulin levels, blood pressure, and heart rate in overweight men. It was suggested that piceatannol could be successfully included in nutritional supplements to improve insulin sensitivity in obese men [88]. In a clinical trial, 36 HIV patients with lipodystrophic syndrome secondary to antiretroviral therapy consumed 30 g/day of passion fruit peel flour (PFPF) with diet therapy for 90 days. The use of passion fruit peel flour effectively reduced total cholesterol as well as triacylglycerides and increased HDL cholesterol [89].

### 4.4. Hepatoprotective Activities

*P. edulis* leaves showed hepatoprotective activity against CCl_4_-induced hepatotoxicity in rats. The ethanol extract from the leaves significantly reduced the levels of serum markers of liver damage, such as alanine transaminase (ALT), aspartate transaminase (AST), and alkaline phosphatase (ALP). It also reduced the levels of oxidative stress markers, such as malondialdehyde (MDA), and increased the levels of antioxidant enzymes, such as superoxide dismutase (SOD) and catalase (CAT) [90]. The hepatoprotective and nephroprotective activities of peel extracts from three varieties of *Passiflora* species (*Passiflora verrucifera* Lindl., *P. edulis*, and *P. ligularis*) administered to albino rats have been examined. All three extracts demonstrated hepatoprotection and nephroprotection in a dose-dependent manner; the most potent was the activity of *P. edulis* peel extract [70]. *P. edulis* peel flour also has a beneficial effect in a model of non-alcoholic fatty liver disease in rats, where it reduces markers of liver damage as well as oxidative stress and improves liver histology [91]. *P. edulis* peel flour (PEPF) supplementation for 8 weeks prevented insulin resistance, hepatic steatosis, and adiposity induced by a low-fructose diet in young rats [92].

### 4.5. Antibacterial and Antifungal Activity

Water extract of *P. edulis*, 8 mg/mL, has shown antibacterial activity against *Staphylococcus aureus*, *Listeria monocytogenes*, *Escherichia coli*, *Enterobacter cloacae*, *Salmonella typhimurium*, and various fungi (*Aspergillus fumigatus*, *A. versicolor*, *A. niger*, *Penicillium funiculosum*, *P. ochloron*, and *Trichoderma viride*) [71]. A study examined the antibacterial activity of *P. edulis* seed extract against *Propionibacterium acnes*, associated with acne. An inhibitory effect on *P. acnes* growth was observed, comparable to antibiotics such as clindamycin and erythromycin [69]. Antibacterial properties of *Passiflora foetida* L. leaf and fruit ethanol and acetone extracts were demonstrated against four pathogenic bacteria: *Pseudomonas putida*, *Vibrio cholerae*, *Shigella flexneri*, and *Streptococcus pyogenes*. The ethanol extract showed a broader spectrum of antibacterial activity, while the acetone extract was most potent against *V. cholerae*. The leaf extracts exhibited better antibacterial activity than the fruits [93]. Antimicrobial proteins from *P. edulis* inhibited the growth of yeasts (*Kluyveromyces marxiannus*, *Candida albicans*, and *Candida parapsilosis*) and impaired the fungal metabolism of *Saccharomyces cerevisiae* as well as *C. albicans*. Some results suggest that 2S albumins might serve as targets for designing new antifungal drugs [94,95].

### 4.6. Anti-Inflammatory and Antipyretic Activities

Numerous reports indicate the anti-inflammatory activity of *Passiflora* species. A study evaluated the inhibitory effects of fruit extracts from *P. edulis* f. *flavicarpa*, *P. edulis*, and *P. ligularis* on inflammation-induced intestinal barrier dysfunction in Caco-2 cells. [28]. Acetone extracts of *Passiflora subpeltata*, administered orally, demonstrated dose-dependent anti-inflammatory effects on a carrageenan-induced paw edema model in rats [96]. Ethanol extract of *P. foetida* leaves at a dose of 100 mg/kg exhibited a significant anti-inflammatory effect in two rat models—carrageenan-induced and histamine-induced paw edema [97].

Another study demonstrated the anti-inflammatory effects of *Passiflora* ethanolic extract upon intragastric administration in rats, at doses ranging from 75 to 500 mg/kg [98]. *P. edulis* f. *flavicarpa* aqueous leaf extract, administered intraperitoneally, exerted anti-inflammatory effects, inhibiting leukocytes, neutrophils, myeloperoxidase, nitric oxide, TNF-alpha, and IL-1 beta levels in carrageenan-induced pleurisy [99]. Aqueous extract of *P. edulis* leaves showed antioxidant and anti-inflammatory effects on a 2,4,6-trinitrobenzene sulphonic acid-induced colitis model, improving antioxidant status and decreasing lipid peroxidation in serum, the liver, and the colon [100]. Acetone extracts of *Passiflora leschenaultii* DC. leaves exhibited acute and chronic anti-inflammatory activity in carrageenan and cotton-pellet-induced rat models. Leaf extracts, at a dose of 400 mg/kg (PO), reduced pain, inflammation, and showed antipyretic effects comparable to those of paracetamol [101].

### 4.7. Antitumor Activity

*P. alata* leaf extract showed cytotoxic potential against cancer cell lines [102]. Antitumor activity is exhibited by extracts of *P. edulis* f. *flavicarpa* in both in vitro and in vivo models. A study explored the effects of leaf and juice extracts *of P. edulis* f. *flavicarpa* on liver cancer cells (HepG2); the leaf extract had a higher polyphenolic content, while the juice extract contained more polysaccharides. The juice extract at 400 μg/mL reduced cell viability, while the leaf extract at 25 μg/mL increased cytotoxicity, and both extracts enhanced proapoptotic activity [103]. Extracts of *P. edulis* f. *flavicarpa* were examined for in vitro cytotoxicity in MCF-7 cells, as well as for in vivo antitumor activity in male Balb/c mice inoculated with Ehrlich carcinoma cells. The fluid extract exhibited higher antitumor activity compared to the crude extract, resulting in a 48.5% inhibition of tumor growth in Balb/c mice and increased lifespan by approximately 42% [104].

### 4.8. Cardiovascular Effects

Experimental data reveal the therapeutic potential of *Passiflora* species in cardiovascular diseases. An ethanolic extract of *P. edulis* leaves was found to reduce blood pressure in hypertensive rats, induce the relaxation of aortic rings, decrease heart rate, and inhibit calcium influx in vascular smooth muscle cells. These effects may be attributed to the presence of flavonoids in *Passiflora* extract [105]. In another study, an ethanol extract of *Passiflora* was demonstrated to increase the force of cardiac muscle contraction in animal models without affecting blood pressure [106]. Additionally, the peel extract of *P. edulis* reduced serum nitric oxide levels, blood pressure, and hemodynamic parameters in spontaneously hypertensive rats [107,108]. The vascular benefits are attributed to compounds like piceatannol in *P. edulis* seeds [109]. A study investigated the effect of the consumption of *Passiflora setacea* DC. juice on inflammation, metabolic parameters, and gene expression in circulating immune cells in humans. The analysis revealed 1327 differentially expressed genes after juice consumption, including those involved in inflammation, cell adhesion, and cytokine–cytokine receptor processes. These results suggest that *P. setacea* consumption may help prevent cardiometabolic diseases [110].

### 4.9. Application in Cosmetics

Aqueous and ethyl acetate extracts from the seed residue of *Passiflora* juice processing exhibited potent antioxidant activity. The ethyl acetate extract also demonstrated ferric-reducing power and a tyrosinase inhibitory effect. The recovered antioxidant fraction holds promise as a sunscreen and skin-lightening agent, making it suitable for inclusion in anti-aging products due to its antioxidant activity and UV protection properties [111]. Additionally, an aromatic oil obtained from residues during passion fruit processing showed a high saponification index, making it suitable for use in manufacturing soaps, shampoos, and cleaning products [112]. *P. edulis* seed extract was found to stimulate collagen production and inhibit elastin degradation, preserving skin moisture and elasticity [113]. Lipid nanoparticles containing passion fruit seed oil as a liquid lipid and glyceryl distearate as a solid lipid were deemed suitable for skin administration and could be incorporated into semi-solid formulations to enhance their skin applications [114].

## 5. Adverse Reactions and Clinical Side Effects

Side effects associated with *Passiflora* flower extracts have been rarely reported and may include allergic reactions, sinus irritation, rhinitis, and skin rashes [115,116]; however, it is believed that these reactions are more likely due to various other components that combine with *Passiflora* preparations. One case reported nausea, vomiting, drowsiness, and tachycardia in an individual after taking *Passiflora* [117]. Due to its sedative effects, *Passiflora* may cause drowsiness and increase reaction time, so caution is advised when driving or operating heavy machinery. There is also a theoretical risk of increased clotting time. While cases of liver toxicity related to the use of kava kava (*Piper methysticum* G.Forst.) have been identified, not many cases related to *Passiflora* toxicity have been reported; however, there is one report of a patient who died from liver failure after consuming a product containing both kava and *P. incarnata.* This suggests a potential synergistic effect between these substances, leading to severe outcomes [118].

### 5.1. Pharmacology, Pharmacokinetics, Safety, and Toxicity

A benzoflavone (BZF) was identified in the aerial parts of *P. incarnata*. BZF, when administered concurrently with the cannabinoid delta 9-THC to mice, prevented the development of dependence and tolerance to cannabinoids [119]. Preliminary studies also evaluated the use of BZF from *P. incarnata* for treating nicotine addiction, where mice administered with combinations of nicotine and BZF exhibited reduced withdrawal effects compared to those treated with nicotine alone [120]. The comparison of the effects of aqueous and ethanolic extracts of the aerial parts of *P. incarnata* showed that the action of the aqueous extract was more potent, even at low doses [121]. *P. edulis* is generally considered non-toxic. In vivo studies showed that an ethanolic extract at a dose of 550 mg/kg had no toxic effects on rats. Aqueous leaf extracts, even at doses of 2000 mg/kg, were found to be safe [121]; however, some reports suggest the potential toxicity of aqueous extract of *P. incarnata* in mice at a dose of 900 mg/kg body weight [121]. The median lethal dose (LD50) for 30% ethanolic extracts taken orally is reported as 37 mL/kg, and, for ethanolic extracts, it is 15 mg/kg. [122]. There is no evidence of genotoxicity in the literature; however, caution is advised for nursing mothers and children when consuming *P. incarnata* extracts, and this should be carried out under medical supervision. *P. incarnata* extracts have been shown to stimulate uterine contractions in animal models [121], so safety during pregnancy and lactation has not been established. The concomitant administration of *P. incarnata* extracts and sedatives, such as benzodiazepines, zolpidem, or alcohol, is not advisable, as pharmacokinetic interactions may occur [123].

### 5.2. Synergic Effects

The absence of proven toxicity allows for the development of nutritional supplements with *Passiflora* without negative health effects. Researchers are exploring synergistic effects between *Passiflora* and other herbal medicinal plants, as well as possible antagonistic interactions between isolated substances and those contained in *Passiflora* extracts. Combining *Passiflora* plant extracts with antibiotics has been found to alter natural antimicrobial resistance, potentially enhancing antibiotic activity. A supplement containing *Passiflora* extract in combination with an antibiotic was reported to increase the antibiotic’s activity [124,125]. In a comparison between standardized extracts of kava kava, *Passiflora*, and their combination, the sedative and hypnotic effects were 50% higher for the combination compared to the effects of the individual extracts used separately [126]. *Passiflora* extract, in combination with *Hypericum* extract with low hyperforin content, demonstrated similar efficacy in treating mild depressive states as mono preparations with high hyperforin content [127]. Fiebich et al. demonstrated the synergistic effects of *Passiflora* extract when administered in combination with a *Hypericum* extract using antidepressant pharmacological models. *Passiflora* significantly enhanced the pharmacological potency of *Hypericum*. The antidepressant effect of *Hypericum* and *Passiflora* extract at specified doses was more potent than that of *Hypericum* extract alone or imipramine [128]. The synergistic effect of *Passiflora* and *Hypericum* on serotonin reuptake can be explained by the altered coupling of phosphorus in the transport molecule of the extracts and inhibition of protein kinases or phosphatases, or by the alteration of protein–protein interactions due to conformational changes of the extract molecule after binding to other molecules [129]. A similar effect has been reported when two antidepressants, escitalopram and R-citalopram, bind to the serotonin transporter [130].

## 6. Patented Products Containing *Passiflora*, Different from the Widely Available Nutritional Supplements Containing *Passiflora* Extracts

Several patented products involving *Passiflora* have been developed:Aromatic capsules with chitosan and *P. edulis* peel extract were designed to reduce lipid and carbohydrate absorption, making them suitable for use in overweight individuals [131].A dermatological complex against wrinkles and skin aging was developed, which includes extracts from *Papaver*, *Metha*, and *Myrtus*, as well as seeds of *P. edulis*, *P. incarnata*, and *P. laurifolia* L. The product was tested on 18 patients with dry skin and resulted in a 41% increase in skin moisture compared to the use of the same dermatological complex without the plant extracts [132].An antidepressant product containing L-glycine, L-methylfolate, magnesium L-threonate, and *P. incarnata* extract was patented. This product was shown to increase the swimming and climbing times of mice in antidepressant models compared to those of controls [133].A product designed to improve the condition of migraine patients was patented, involving a liquid extract of *P. incarnata* obtained in water–ethanol mixtures with an ethanol concentration of 30–70%. After three weeks of intake, most patients reported a reduction in the frequency, strength, and duration of migraine attacks [134]. According to Lorente et al., the general condition of patients improved by over 89% [135].Pharmaceutical formulations with antiulcer activity were developed, utilizing gelatin nanoparticles with *P. alata* dry leaf and stem extract (2–5%) in addition to gelatin (95–98%) [135].

Information on some of the main effects and recommended daily doses are presented in Table 5.

## 7. Conclusions

*Passiflora* is used most often in the food industry; it contains high levels of essential elements, essential amino acids, saccharides, and vitamins, and has a high potential for applications in various nutritional supplements, cosmetics, and pharmaceutical products. All parts of *Passiflora* (fruit, leaves, and stems) contain the listed substances and biologically active components. Over the years, much scientific evidence has been accumulated and several clinical studies have been conducted that support the use of *Passiflora* in various aspects affecting human health. Possible toxic and synergistic effects have been evaluated, and attention has been paid to its application. The use of *Passiflora* is expanding with the development of cosmetic preparations, and in recent years various products containing *Passiflora* have even been patented.

## Figures and Tables

**Figure 1 plants-13-00228-f001:**
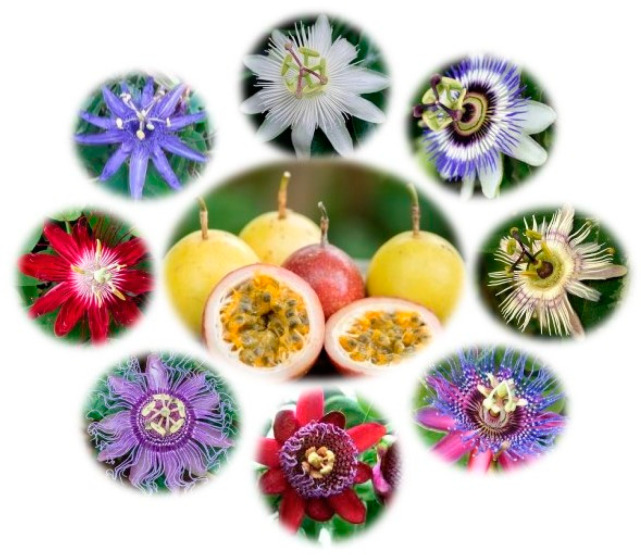
Flowers and fruits of species of *Passiflora*.

**Figure 2 plants-13-00228-f002:**
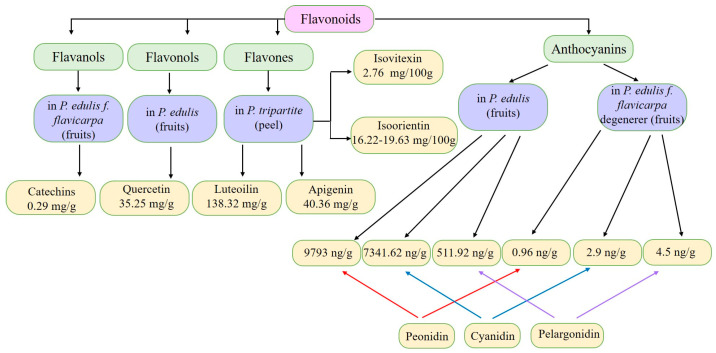
Flavonoid content of *Passiflora* fruit and peel [28,29,30,31,32,33,34].

**Figure 3 plants-13-00228-f003:**
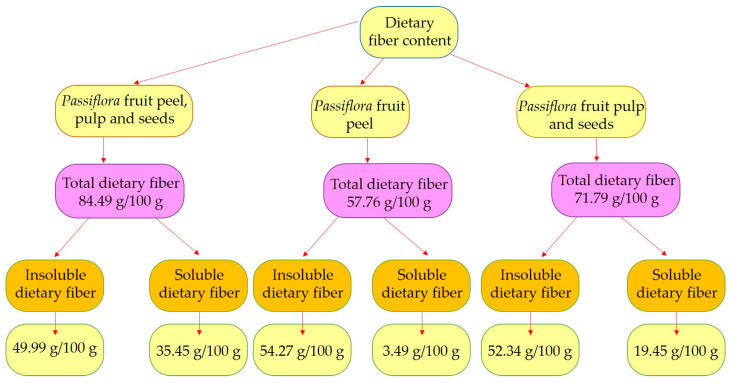
Soluble and insoluble fibers in different parts of *Passiflora* species [61,62,63].

**Figure 4 plants-13-00228-f004:**
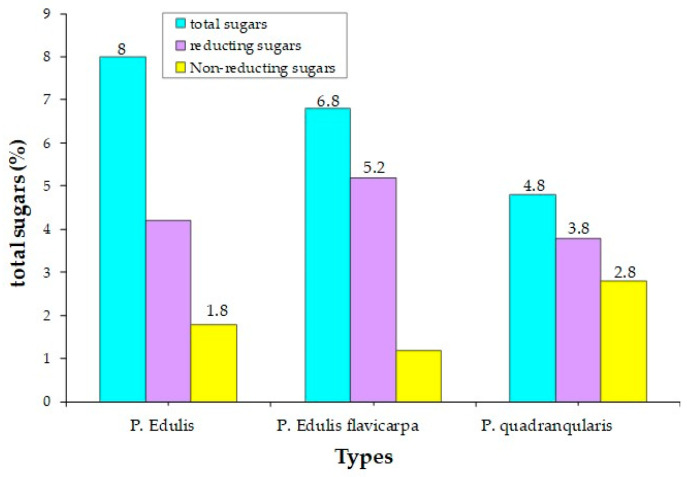
Sugar content of the ripe fruits of some *Passiflora* species.

**Figure 5 plants-13-00228-f005:**
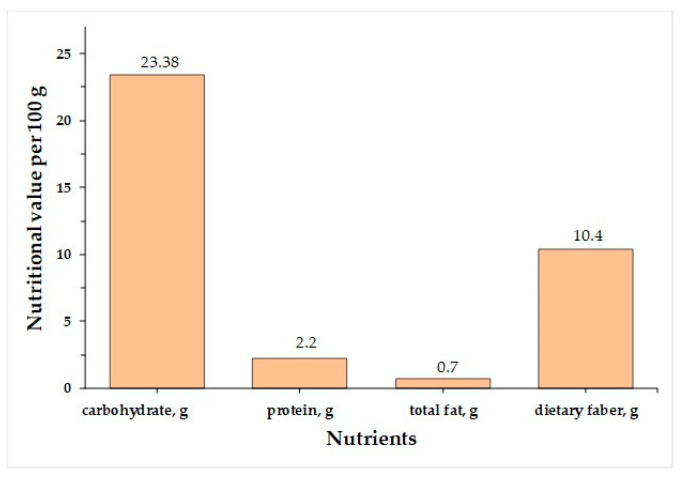
Major nutrients in the fruits of some *Passiflora* species.

**Figure 6 plants-13-00228-f006:**
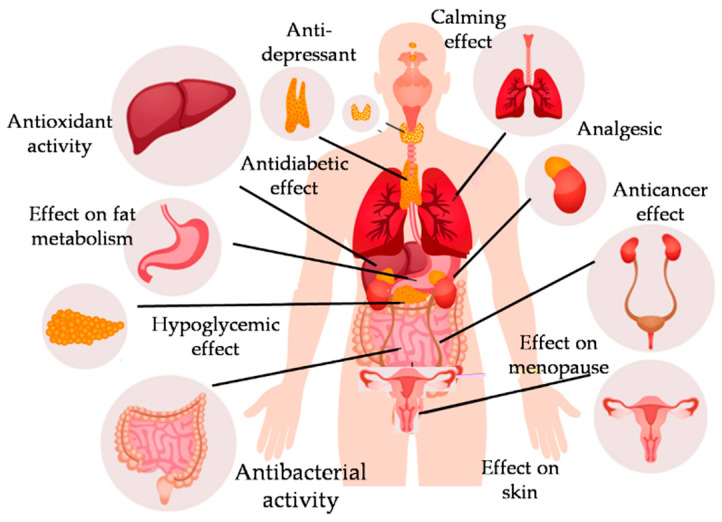
Health benefits of Passiflora.

**Table 1 plants-13-00228-t001:** Distribution of plants and fruits description.

Species	Habitat/Distribution	Characteristics of Fruits	References
*Passiflora edulis* Sims (purple passion fruit)	South America and Mexico Adapted for cultivation in China	The fruit is round, purple, 4–7 cm in diameter, and 4–9 cm long, weighing about 35–45 g	[4]
*Passiflora edulis* Sims *f. flavicarpa* O. Deg. (yellow passion fruit)	A mutation of the purple variety or natural hybrid. Commonly found in Northeast or South India [5]	The fruit is round, golden yellow, 4–7 cm in diameter, and 6–12 cm long, weighing about 60 g	[4,5,6]
*Passiflora quadrangularis* L. (giant granadilla)	The tropical regions of America and Asia	The fruit is greenish-yellow, 15–20 cm long, and weighs about 600 g	[4]
‘Kaveri’ Hybrid passion fruit (purple and yellow)	A hybrid, developed in India by crossing purple and yellow passion fruit	The fruit is purple, weighing about 85–110 g	[4]

**Table 2 plants-13-00228-t002:** Chemical constituents of the *Passiflora* genus (some classes of metabolites).

Chemical Compound	Species	Part of the Plant	References
Citric acid	*Passiflora mollissima* L.H. Bailey	Fruit pulp	[35]
Syringic acid	*Passiflora mollissima* L.H. Bailey	Fruit pulp	[35]
Caffeic acid	*Passiflora subpeltata Ortega*	Leaves	[36]
Caffeoylquinic acid	*Passiflora tenuifila* Killip	Fruits	[37]
Malic acid	*Passiflora edulis* Sims, *Passiflora edulis* f. *flavicarpa* O. Deg, *Passiflora alata* Curtis	Fruits	[38]
Apigenin-C-hexoside-C-pentoside	*Passiflora foetida* L.	Aerial parts (as mixed)	[39]
Vitexin (Apigenin-8-C-glucoside)	*Passiflora foetida* L.	Aerial parts (as mixed)	[39]
Luteolin-C-hexoside-C-pentoside	*Passiflora foetida* L.	Aerial parts (as mixed)	[39]
Luteolin	*Passiflora edulis* Sims, *Passiflora edulis* f. *flavicarpa* O. Deg	Fruit peel fruit	[29]
Quercetin	*Passiflora edulis* Sims, *Passiflora edulis* f. *flavicarpa* O. Deg	Fruit peel fruit	[29]
Naringenin	*Passiflora mollissima* L.H. Bailey	Fruit pulp	[35]
Trihydroxy(iso)flavanol-(epi)catechin	*Passiflora mollissima* L.H. Bailey	Fruit pulp	[35]
Tyrosine	*Passiflora edulis* Sims (cultivar ‘Tainung No. 1’)	Seed	[40]
Valine	*Passiflora edulis* Sims (cultivar ‘Tainung No. 1’)	Seed	[40]
Proline	*Passiflora edulis* Sims (cultivar ‘Tainung No. 1’)	Seed	[40]
Glycine	*Passiflora edulis* Sims (cultivar ‘Tainung No. 1’)	Seed	[40]
α-Humulene	*Passiflora sexocellata* Schltdl., *Passiflora trifasciata* Lem.	Leaves, flowers	[41]
Pentadecanoic acid	*Passiflora sexocellata* Schltdl. *Passiflora trifasciata* Lem.	Leaves, flowers	[41]
Methyl salicylate	*Passiflora trifasciata* Lem.	Leaves, flowers	[41]
Benzyl alcohol	*Passiflora sexocellata* Schltdl., *Passiflora trifasciata* Lem.	Leaves, flowers	[41]
Cyclooctasiloxane, Hexadecamethyl	*Passiflora edulis* Sims	Fruit pulp	[42]
Cyclononasiloxane, Octadecamethyl	*Passiflora edulis* Sims	Fruit pulp	[42]

**Table 3 plants-13-00228-t003:** Vitamin compositions of juices prepared from the fruits of *P. edulis* and *P. edulis* f. *flavicarpa* [60].

Type of Vitamin, Units	Juice from *P. edulis* f. *flavicarpa*	Juice from *P. edulis*
Vitamin B6, mg	0.06	0.05
Vitamin A RAE, μg	47	36
Vitamin A, IU	943	717
Vitamin E (alpha tocopherol), mg	0.01	0.01
Vitamin K, μg	0.4	0.4

**Table 4 plants-13-00228-t004:** Chemical compositions in *Passiflora* and pharmacological effects.

Chemical Formula	Effect	Reference
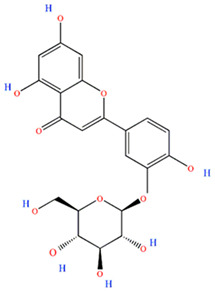 Luteolin-3-glucoside	Antitumor	[21]
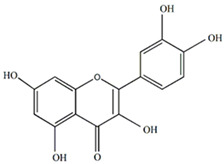 Quercetin	Antiallergic, antitumor, antiviral, and neuroprotective	[21,43,44,45]
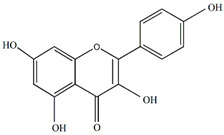 Kaempferol	Antiallergic, antitumor, antiviral, and neuroprotective	[21,43,44,45]
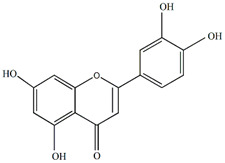 Luteolin	Antiallergic, antitumor, antiviral, and neuroprotective	[43,44,45]
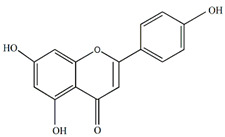 Apigenin	Antiallergic, antitumor, antiviral, and neuroprotective	[21,43,44,45]
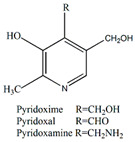 Vitamin B_6_	Antiviral, immunological, and neuroprotective	[59]
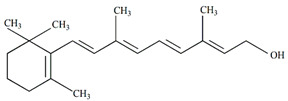 Vitamin A	Antiviral, immunological, and neuroprotective	[59]
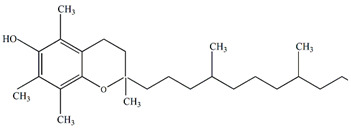 Vitamin E	Antiviral, immunological, and neuroprotective	[59]
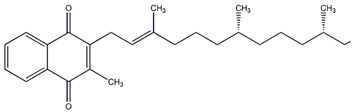 Vitamin K	Antiviral, immunological	[59]
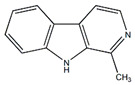 Harmane	Anti-inflammatory, antitumor	[59]
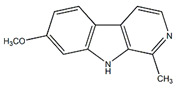 Harmine	Anti-inflammatory, antitumor	[4]
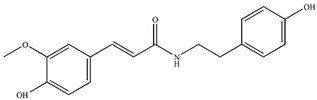 N-trans-feruloyltyramine	Anti-inflammatory, antitumor	[4]
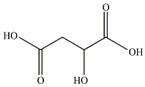 Malic acids	Antioxidant	[4]
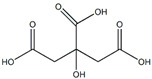 Citric acids	Antioxidant	[4]

**Table 5 plants-13-00228-t005:** Main applications of *Passiflora* species in medicine.

Passiflora Species	Ailment	Dose	Ref.
*P. incarnata*	Dental anxiety	20 drops/day	[80]
		260 mg	[136]
*P. incarnata*	Preoperative anxiety	1000 mg	[137]
		500 mg	[138]
		700 mg/5 mL	[79]
*P. incarnata*	Anxiety disorder	45 drops/day	[77]
*P. incarnata*	Sleep disorder	infusion (2 g in 250 mL)	[81]
*P. incarnata*	Attention-deficit hyperactivity disorder (ADHD)	0.04 mg/kg/day (twice daily)	[139]
*P. incarnata*	Opiate detoxification	60 drops/day + 0.8 mg of clonidine	[140]

## Data Availability

Datasets from the time of this study are available from the respective authors upon reasonable request.

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
