# Peer review of "Chemical Compositions, Pharmacological Properties and Medicinal Effects of Genus Passiflora L.: A Review"

_plants, 2024, doi:10.3390/plants13020228_

Round 1
Reviewer 1 Report
Comments and Suggestions for Authors
The study provides scientifically important data regarding the applications and safety of dietary supplements, medicines, and cosmetics based on different herbal preparations from Passiflora plants.
Comments
-The authors should correct the sentence in lines 24-26. Antibacterial effects are mentioned twice.
- Some parts of the manuscript contain unnecessary text in brackets (for example, in lines 34, 35, 45, 103, etc.)
- The subheadings in the section “Medical effects and activities in pharmacy” and their content should be organized more precisely. For example, the subheading “4.1. Pharmacological activities: antibacterial, hepatoprotective, nephroprotective” does not adequately describe the following content. In addition, the authors described the antibacterial effects in lines 249-266
-Line 165-168. Please check the statements and correct the references provided. Statements appear not adequately referenced; the references 59 and 60 are the same.
- Please check references throughout the manuscript
- Figure 5 should be corrected. The “Hypoclycemic” should be corrected into “Hypoglycemic”; “Health Benefits” are mentioned twice.
Comments on the Quality of English Language- Minor editing of English language required
Author Response
Dear reviewer,
We attached the answers of your questions in the file below.

Reviewer 2 Report
Comments and Suggestions for Authors
I'm glad to see this article titled "Chemical composition, pharmacological properties and medicinal effects of genus Passiflora L." This is a rather classic review article focusing on a plant genus. However, considering the content, I would suggest submitting this article to a journal specializing in food-related topics. Here are some personal suggestions and queries I have:
1.I suggest using the standardized plant names according to the WFO Plant List(https://wfoplantlist.org/plant-list/). For instance, Passiflora edulis Sims f. edulis should be written as Passiflora edulis Sims, and Passiflora edulis Sims f. flavicarpa should be Passiflora edulis f. flavicarpa O.Deg.
2.The plant classification system of angiosperms in Section 2 doesn’t need to be presented.
3.If you aim to systematically review the Passiflora genus, it would be preferable to emphasize the genus name more and avoid using "Passion flower plant" (Line 57).
4.Regarding Line 67, does the chemical composition of “Passion flowers” possess these pharmacological effects? Please add cited references.
5.In Line 81, when you mention "yellow skin and purple skin fruits in all Passiflora genus," are you referring to fruits across the entire Passiflora genus or specific varieties within the genus? Just as you mentioned in your text that there are 16 genera and over 600 species in the Passifloraceae family. It's advisable to describe your content using specific plant names wherever possible.
6.You could consider creating a table for the compounds' structures and their corresponding pharmacological effects to enrich the content of your article.
7.The content in Section 3 about chemical composition predominantly showcases the value of Passiflora genus fruits as everyday food items.
8.Lines 113-116 refer to the discovery of trace elements in Passiflora alata in 2021. Is it the discovery of trace elements or the enhancement of immunity due to trace elements in 2021? If it's the former, the 45th reference doesn't mention this; please cite the new reference. If it's the latter, consider rephrasing it in a less disputable manner.
9..Please confirm if the citation formats for references 50 and 51 are correct.
10. In Figure 5, do the different colors of the hexagon represent a deeper meaning? Please provide a legend in the diagram to explain your classification. Also, why are there two "health benefits"?
11.Compared to the full text, the content of the Section 7 seems a little short.
Comments on the Quality of English LanguageMinor editing in English is necessary.
Author Response
Dear reviewer,
We attached the answers in the letter below.

Reviewer 3 Report
Comments and Suggestions for Authors
The authors review the nutritional and pharmacological benefits of passion fruit plant components as conveyed through many tens of published studies. The review is extensively referenced and, although somewhat superficial in its treatment, does alert readers to sources of more detailed information. The scope of the review is quite broad, but that seems to be its intent.
Although very readable and generally well-written, there are a few corrections that should be made:
· Why are statements underlined and in ( ) in the first paragraph and in lines 45-46?
· There are no column headings in Table 1, columns 3 and 4
· The statement in line 88 is not necessarily correct. BBB permeability may ‘positively affect’ the beneficial properties, but it does not “explain” them, because pharmacokinetics does not define the mechanism of action.
· The review is filled with one sentence paragraphs. These should be combined when related to a topic sentence to from complete (multi-sentence) paragraphs.
· The column 2 column width is too small in Table 2 (affects word wrapping)
· “(“ should be removed to start line 103
· “N-trans-“ and “cis-N-“ in line 146 must be italicized.
· Clinical studies, when mentioned (e.g., lines 172, 180 and others), should be more descriptive. For instance, what was the composition of the cohort? What was the end point?
· What is meant by “sexual problems” in line 179? ED, loss of libido, etc. More specificity would be helpful.
· “benefisial” is a misspelling in line 190.
· Remove “the” at the end of line 225 and in line 247
· Add ‘a’ before “low-fructose” in line 247 to improve sentence clarity.
· “Hypoclycemic” is a misspelling in Figure 5.
· In Figure 6, the figure should read ‘2.5 per day’ rather than “2,5 for day”
Despite the few errors, I recommend publication after the items listed are addressed by the authors.
Comments on the Quality of English LanguageGenerally fine, with the few exceptions noted in the 'Suggestions for Authors'.
Author Response
Dear reviewer,
I attached the answers of your questions.

Reviewer 4 Report
Comments and Suggestions for Authors
Dear Editor,
the manuscript is not well organized and comprehensively described, the relative content of each paragraph should be more deeply argumented, as requested by a review article. Morever, very recently a comprehensive review on the same topic has been published (https://doi.org/10.1016/j.foodchem.2023.136825), thus the manuscript also lacks in novelty. Therefore, I recommend not to publish it.
Comments on the Quality of English LanguageModerate editing of English language required
Author Response
Dear Editor,
the manuscript is not well organized and comprehensively described, the relative content of each paragraph should be more deeply argumented, as requested by a review article. Morever, very recently a comprehensive review on the same topic has been published (https://doi.org/10.1016/j.foodchem.2023.136825), thus the manuscript also lacks in novelty. Therefore, I recommend not to publish it.
We have introduced corrections to the text on the comments made by the reviewers. The already published overview only confirms the actuality of the topic. We believe that each review has a contribution to the needs of researchers and should not be easily dismissed.
Round 2
Reviewer 2 Report
Comments and Suggestions for Authors
I'm pleased to see the revisions you've made. The pharmacological section in the latter part of the article is very well done. However, there are still some issues in the taxonomy and chemical composition sections.
1.Firstly, the taxonomy section contains non-taxonomical content, especially in the last two paragraphs. This seems more suitable for the introduction section and I suggest merging it there. It might be helpful to include some plant images to better align with the aesthetics of this journal.
2.Secondly, in describing the chemical composition, it would be beneficial to follow the format used in recent reviews published in our target journal. While visual aids might be a novel approach, I recommend adding tables, or you could refer to the style used in https://doi.org/10.3390/plants12223795.
3.Moreover, there seem to be issues with the chemical bonds in the chemical structures in Table 3. Consider using 'ChemDraw' software to accurately depict these chemical formulas.
4.Lastly, regarding the visual coloring of images, I suggest referencing the article DOI: 10.1038/s41467-020-19160-7 or other articles within the same journal, such as https://doi.org/10.3390/plants12152860. These can provide useful guidance.
Author Response
I'm pleased to see the revisions you've made. The pharmacological section in the latter part of the article is very well done. However, there are still some issues in the taxonomy and chemical composition sections.
- Firstly, the taxonomy section contains non-taxonomical content, especially in the last two paragraphs. This seems more suitable for the introduction section, and I suggest merging it there. It might be helpful to include some plant images to better align with the aesthetics of this journal.
The last two paragraphs are appended to the introduction. A figure representing some flowers and fruits of Passiflora is also included.
- Secondly, in describing the chemical composition, it would be beneficial to follow the format used in recent reviews published in our target journal. While visual aids might be a novel approach, I recommend adding tables, or you could refer to the style used in https://doi.org/10.3390/plants12223795.
Added Table 2 contains information on the chemical composition of the genus Passiflora.
- Moreover, there seem to be issues with the chemical bonds in the chemical structures in Table 3. Consider using 'ChemDraw software to accurately depict these chemical formulas.
As you suggested, the formulas in Table 4 were made using the 'ChemDraw' software.
- Lastly, regarding the visual coloring of images, I suggest referencing the article DOI: 10.1038/s41467-020-19160-7 or other articles within the same journal, such as https://doi.org/10.3390/plants12152860. These can provide useful guidance.
We have replaced Figure 6 with the new one.
Reviewer 4 Report
Comments and Suggestions for Authors
Dear Editor,
the manuscript in its second revision is quite improved but I think that to gain the publication it must be revised again, trying to give to the reader immeadiate and easy-readible information about medical properties of the plant. That is why other works are available in literature and this one might be unique and original by itself. So, I suggest to create a new table summarizing the medical effects exterted by the plants, also reporting the recommended doses of the plant extract and the related references.
Comments on the Quality of English Languagemoderate editing of english language
Author Response
Dear Editor,
the manuscript in its second revision is quite improved but I think that to gain the publication it must be revised again, trying to give to the reader immeadiate and easy-readible information about medical properties of the plant. That is why other works are available in literature and this one might be unique and original by itself. So, I suggest to create a new table summarizing the medical effects exterted by the plants, also reporting the recommended doses of the plant extract and the related references.
The recommended table has been added to the text (Table 5).
Round 3
Reviewer 2 Report
Comments and Suggestions for Authors
I'm glad to see your revisions again. The sections on taxonomy and chemical composition are clearer than before, but I have a few points to address:
1.The title of the article has an extra "Title".
2. Regarding Figure 5, I suggest using softer colors to enhance its visual effect.
3. I noticed the addition of Table 5 at the end of the article and have a few queries: Why was the information about the "Main applications of Pasiflora species in medicine" placed within the section on "patented products"? Why does the table contain references (e.g., the newly cited references 140 -143) not mentioned in the main body of the article? Are all the items listed in the table already clinically used products? While the intent behind creating this table is commendable, it might be beneficial to consider what content the readers are most likely seeking for quick reference. Additionally, there is a spelling error for the “Pasiflora” in Table 5, and reference 138 was published in 2016, not 2017.
Lastly, unfortunately, I recently came across an article in Food Chemistry that bears some resemblance to your content and appears to be more comprehensive. Nevertheless, I acknowledge your dedication to the study of Passiflora. I encourage you to emphasize the unique aspects of your work, such as the patents and Table 5, to ensure the value of your contribution stands out.
Author Response
I'm glad to see your revisions again. The sections on taxonomy and chemical composition are clearer than before, but I have a few points to address:
- The title of the article has an extra "Title".
Thank you for the remark. It is deleted.
- Regarding Figure 5, I suggest using softer colors to enhance its visual effect.
The colors in the figure have been corrected.
- I noticed the addition of Table 5 at the end of the article and have a few queries: Why was the information about the "Main applications of Pasiflora species in medicine" placed within the section on "patented products"? Why does the table contain references (e.g., the newly cited references 140 -143) not mentioned in the main body of the article? Are all the items listed in the table already clinically used products? While the intent behind creating this table is commendable, it might be beneficial to consider what content the readers are most likely seeking for quick reference. Additionally, there is a spelling error for the “Pasiflora” in Table 5, and reference 138 was published in 2016, not 2017.
Table 5 has been added according to the requirements and recommendations of another reviewer, and we now have his consent to submit the manuscript in this final form. For this reason, references that were not present in the original version have been added.
The technical corrections have been made.
Lastly, unfortunately, I recently came across an article in Food Chemistry that bears some resemblance to your content and appears to be more comprehensive. Nevertheless, I acknowledge your dedication to the study of Passiflora. I encourage you to emphasize the unique aspects of your work, such as the patents and Table 5, to ensure the value of your contribution stands out.
Reviewer 4 Report
Comments and Suggestions for Authors
manuscript deeply revised. It can be published in its current form
Comments on the Quality of English Languageminor editing of english language required
Author Response
Thank you very much for your decision.